# Innovative monitoring scheme adapted to remote, scattered nesting aggregation reveals a major loggerhead turtle rookery in New Caledonia, South Pacific

**Hugo Bourgogne**[1,2]*, **Marc Oremus**[2], **Morgan Mangeas**[1], **Eric Vidal**[1], **Marc Girondot**[3]

**1** UMR 250 ENTROPIE, Centre IRD de Nouméa, IRD, CNRS, Ifremer, Université de la Réunion, Université de la Nouvelle-Calédonie, Nouméa, New Caledonia, France, **2** WWF-France Bureau Nouvelle-Calédonie, World Wide Fund for Nature (WWF), Nouméa, New Caledonia, France, **3** Laboratoire Ecologie, Systématique et Evolution, AgroParisTech, Centre National de la Recherche Scientifique (CNRS), Université Paris-Saclay, Gif-sur-Yvette, France

* hugo.bourgogne@ird.fr

**Data Availability Statement:** All relevant data are within the manuscript and its Supporting information files.

## Abstract

The loggerhead turtle *Caretta caretta* is a large marine turtle with a cosmopolitan repartition in warm and temperate waters of the planet. The South Pacific subpopulation is classified as 'Critically Endangered' on the IUCN Red List, based on the estimated demographic decline. This precarious situation engages an urgent need to monitor nesting populations in order to highlight conservation priorities and to ensure their efficiency over time. New Caledonia encompasses a large number of micro and distant nesting sites, localized on coral islets widely distributed across its large lagoon. Adequately surveying nesting activities on those hard-to-reach beaches can prove to be challenging. As a result, important knowledge gaps prevail in those high-potential nesting habitats. For the first time, an innovative monitoring scheme was conducted to assess the intensity of nesting activities, considered as a proxy of the population size, on an exhaustive set of islets located in the 'Grand Lagon Sud' area. These data were analyzed using a set of statistical methods specially designed to produce phenology and nesting activity estimates using Bayesian methods. This analysis revealed that this rookery hosts a large nesting colony, with a mean annual estimate of 437 nests (95% Credible Interval = 328–582). These numbers exceed that of the previous estimated annual number of loggerhead turtle nests in New Caledonia, highlighting the exceptional nature of this area. Considering the fact that similar high-potential aggregations have been identified in other parts of New Caledonia, but failed to be comprehensively assessed to this day, we recommend carrying out this replicable monitoring scheme to other locations. It could allow a significant re-evaluation of the New Caledonian nesting population importance and, ultimately, of its prevailing responsibility for the protection of this patrimonial yet endangered species.

**Funding:** This study was supported and funded by the French Research Institute for Sustainable Development IRD for salary considerations for HB (through the 'Industrial Agreements for Training through Research' CIFRE scholarship #2020/1154) / EV / MM, and the World Wide Fund for Nature international non-governmental organization WWF-France for salary considerations for HB/MO and operating expenses. The authors declare that no funds, grants, or other supports were received during the preparation of this manuscript.

**Competing interests:** The authors have declared that no competing interests exist.

# 1 Introduction

Assessing population abundance and demographic trends is key to develop and evaluate conservation actions for threatened species [1–3]. Marine megafaunal species, such as sea turtles, face major conservation issues worldwide [4–7]. Yet, long-term monitoring programs can prove to be particularly challenging, due to their ocean-scale distributions and migratory nature [8, 9]. Onshore quantifications of sea turtle nesting activities have developed into the most common indicator to evaluate the demographic trends, as it is often used as an index of population size [9, 10]. Several monitoring schemes have been developed to assess long-term population trends according to nesting site conformations and specificities [11]. Monitoring remote and scattered insular nesting areas with high precision is a statistical challenge. The high financial costs associated with field presence act indeed as a strong hindrance to the collection of core data. This may lead to important knowledge gaps and significant misunderstanding of demographic processes in high-potential habitats [12–14]. The development of efficient and resource-optimized tools is therefore a priority as it could reveal unsuspected major nesting rookeries [13], especially in the South Pacific region that encompasses a multitude of island-rich geographies [15, 16].

The loggerhead turtle *Caretta caretta* (Linnaeus, 1758) is classified as 'Vulnerable' on the International Union for Conservation of Nature's (IUCN) Red List on a global scale. However, the South Pacific subpopulation suffered a strong demographic collapse, showing a loss of over 80% of nesting females over the past three generations. It was therefore classified as 'Critically Endangered' in 2015 [17]. Nesting aggregations of this distinct Regional Management Unit [7] are restricted to the western part of the ocean basin and are reported almost exclusively in eastern Australia and New Caledonia [18, 19]. In Australia, approximately 80% of the loggerhead turtle nesting activities are concentrated at five major areas, each supporting hundreds of nesting females every year [17]. They combine for the greatest concentration of all reproductive individuals in the South Pacific region, as the remainder of the nesting population is dispersed at a large number of small aggregations with 10s or <10 nesting females per year. In New Caledonia, only one important coastal nesting site with several 10s of nesting females every year has been identified on the west coast of the mainland, in the 'Roche Percée' beach, commune of Bourail [20, 21]. Numerous other smaller nesting beaches have been inventoried on dispersed coral islets along the coastline, forming large aggregations off the northwestern coast of the mainland and in the 'Grand Lagon Sud' (GLS) provincial park, at the meridional end of the mainland. Despite the pre identified potential for hosting a significant number of nesting activities, information for those insular rookeries remained sparse and only based on expert opinion because they did not benefit censuses to this date. Given the reported lack of information on the loggerhead turtle demographic status at the national scale [17], the assessment of those nesting areas could prove critical to a re-evaluation of New Caledonia's population and its importance within the South Pacific subpopulation.

In this study, we chose to give a specific focus on the 'GLS' area. It presents strong monitoring logistical constraints, with the requisite coverage of multiple, distant and scattered nesting beaches. We have therefore implemented an exploratory protocol, based on the seasonal survey of 28 islets and supported by an appropriate analytical tool to make up for the scarcity of data generated. Based on a six seasons time-series data collection, we have achieved the two main objectives that were initially set:

1. establish a monitoring protocol well-adapted to the study area, allowing to deliver a baseline knowledge meeting the minimum threshold for data quality [11] while optimizing the resources required for its implementation;

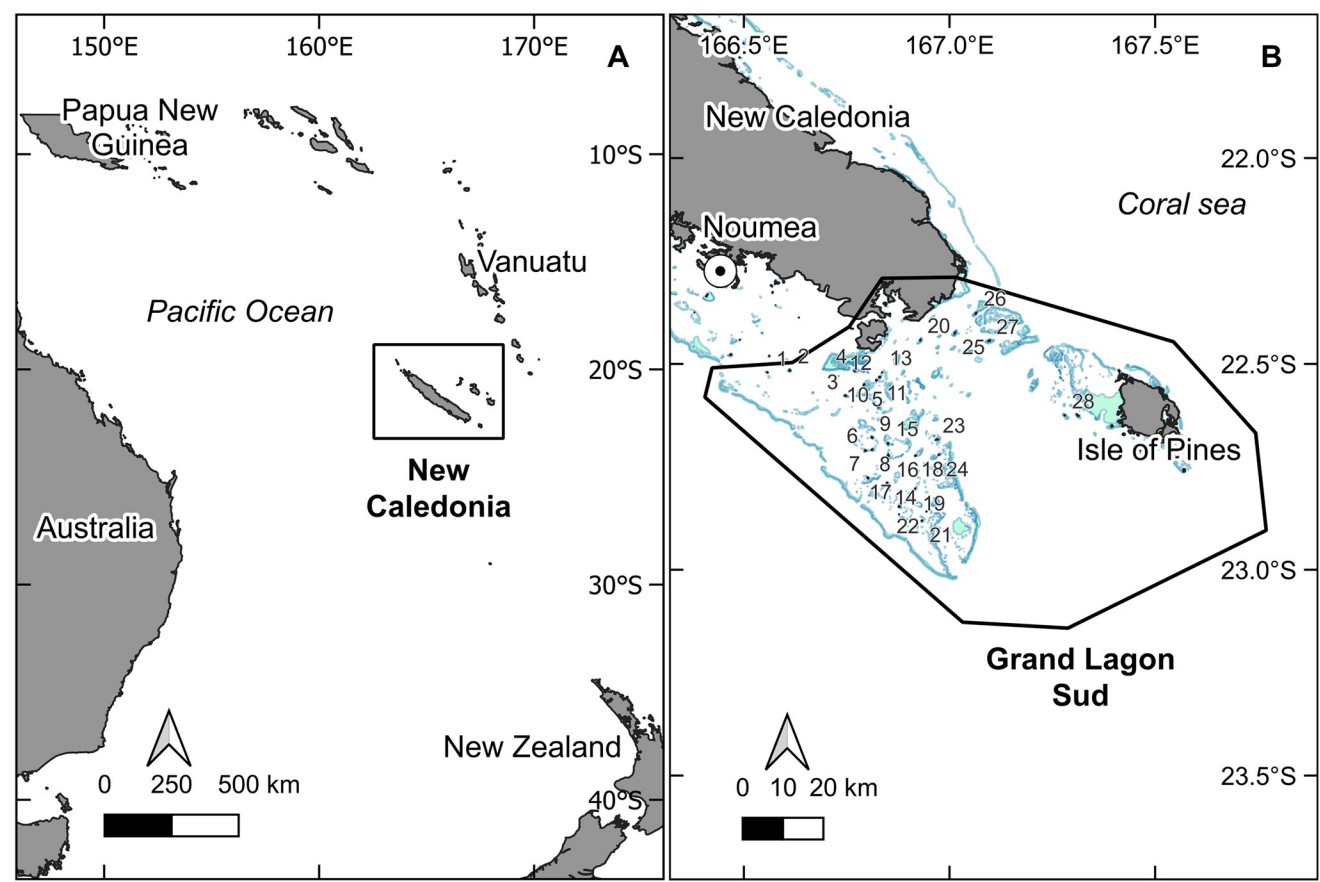

**Fig 1. Geographical location of the study area, in the 'Grand Lagon Sud' of New Caledonia.** A: New Caledonia, South West Pacific. B: Position of the 28 surveyed sites within the 'Grand Lagon Sud' area, off the southern end of the main island of New Caledonia. Land is shown in light grey and shallow reefs are shown in blue (shapefile source: Millenium Coral Reef Mapping Project [22]).

2. produce the first estimates of the annual nesting intensity at the 'GLS' rookery-scale allowing to evaluate the status of this area in relation to the South Pacific loggerhead turtle population.

## 2 Materials and methods

### 2.1 Study area

Censuses were conducted in the 'Grand Lagon Sud' (S22°00, E167°00) of New Caledonia, a 314.500 ha marine protected park in the southern end of the mainland's coral lagoon (Fig 1). Its inscription as a UNESCO world heritage site in 2008 testifies to the exceptional nature of the area and its ecological interest. This region is characterized by a rich coral reef diversity, extensive seagrass and algal communities, a fractioned reef barrier, and 35 distinct coral reef islets distributed over the park.

The surveyed area requires the use of a motorboat to conduct field missions because of the large geographic scope to cover, with 32 km from Noumea city harbor to reach the first monitored islet, and 88 km to reach the most distant islet. All islets present fringing reefs and are

**Table 1. Designation and geographic positions of the 28 surveyed sites in the 'GLS' rookery.**

| islet # | name | latitude | longitude | set | islet # | name | latitude | longitude | set |
|---|---|---|---|---|---|---|---|---|---|
| 1 | atire | -22.5219 | 166.5603 | A | 15 | nge | -22.6926 | 166.8507 | A |
| 2 | redika | -22.5161 | 166.6130 | A | 16 | gi | -22.7212 | 166.8516 | A |
| 3 | vua | -22.5781 | 166.7475 | A | 17 | nda | -22.8456 | 166.8773 | A |
| 4 | uo | -22.5169 | 166.7696 | B | 18 | uie | -22.7208 | 166.9125 | B |
| 5 | mato | -22.5503 | 166.7923 | B | 19 | mbore | -22.8030 | 166.9161 | B |
| 6 | uatio | -22.7090 | 166.7953 | A | 20 | ugo | -22.4417 | 166.9285 | B |
| 7 | kouare | -22.7767 | 166.8009 | A | 21 | koko | -22.8804 | 166.9329 | B |
| 8 | ua | -22.7079 | 166.8112 | A | 22 | petit koko | -22.8584 | 166.9439 | B |
| 9 | uaterembi | -22.6795 | 166.8117 | A | 23 | ndo | -22.6840 | 166.9653 | B |
| 10 | noe | -22.5393 | 166.8218 | B | 24 | totea | -22.7203 | 166.9735 | B |
| 11 | ieroue | -22.6049 | 166.8277 | A | 25 | nouare | -22.4238 | 167.0128 | B |
| 12 | puemba | -22.5321 | 166.8289 | B | 26 | kie | -22.3761 | 167.0646 | B |
| 13 | pumbo | -22.5205 | 166.8355 | B | 27 | amere | -22.4441 | 167.0961 | B |
| 14 | tere | -22.7888 | 166.8475 | A | 28 | du ami | -22.6252 | 167.2797 | B |

The geographic coordinates point the center of the islet.

difficult to access by low tide and/or poor weather conditions, leading to major planning constraints. Given the distance to the eastern part of the park (128 km from the departing harbor to the most eastern islet), the choice was made not to include the 7 islets surrounding the Isle of Pines (Fig 1) to the study area. The remaining 28 coral islets of the area were annually monitored over the 2017–18 to 2022–23 period, for a total of n = 6 nesting seasons covered (Table 1). The islets are stable at the scale of this study.

## 2.2 Monitoring protocol

**2.2.1 Nesting data count type.** Seasonal survey events were conducted during austral summer in the 'GLS' area to match the reproductive phenology of the loggerhead turtle in New Caledonia [20]. They consisted in a series of 1 to 2 days missions at sea, in which a set of islets were monitored by a minimum of two persons in order to reduce observer bias. All nesting activities observed on the perimeter of the islet were recorded, along with date, site, GPS localization and nature of the detected activity based on the available cues: crawls up and down the beach, aborted or successful nest as materialized by the characteristic body pit tracks (S1 and S2 Files). Knowing that only the loggerhead and the Green turtles *Chelonia mydas* (Linnaeus, 1758) are nesting in New Caledonia [23], species identification was assessed through crawling track observations. 99.9% of all tracks were confirmed as asymmetrical, thus assigned to loggerhead turtles [24]. The average perimeter length of the islets is 813.2 m (+ 310.0), and the average duration for a person to complete an islet patrol is 39.5 min (+ 3.4). We chose to use nest counts as the proxy for annual population abundance in this study. The selection of this count unit, rather than the more classical female crawling tracks, has been motivated by our incapacity to visit nesting beaches more than on a weekly basis at best. In this context, the higher persistence in time of body pits compared to crawl tracks makes it a better proxy to avoid detection bias. Moreover, body pits are considered the most accurate proxy to describe patterns in reproductive output after the direct count of individuals [11].

**2.2.2 Monitoring strategy in space and time.** The 'GLS' rookery presents a spatial configuration where numerous, non-contiguous, distant and hard-to-reach nesting beaches are

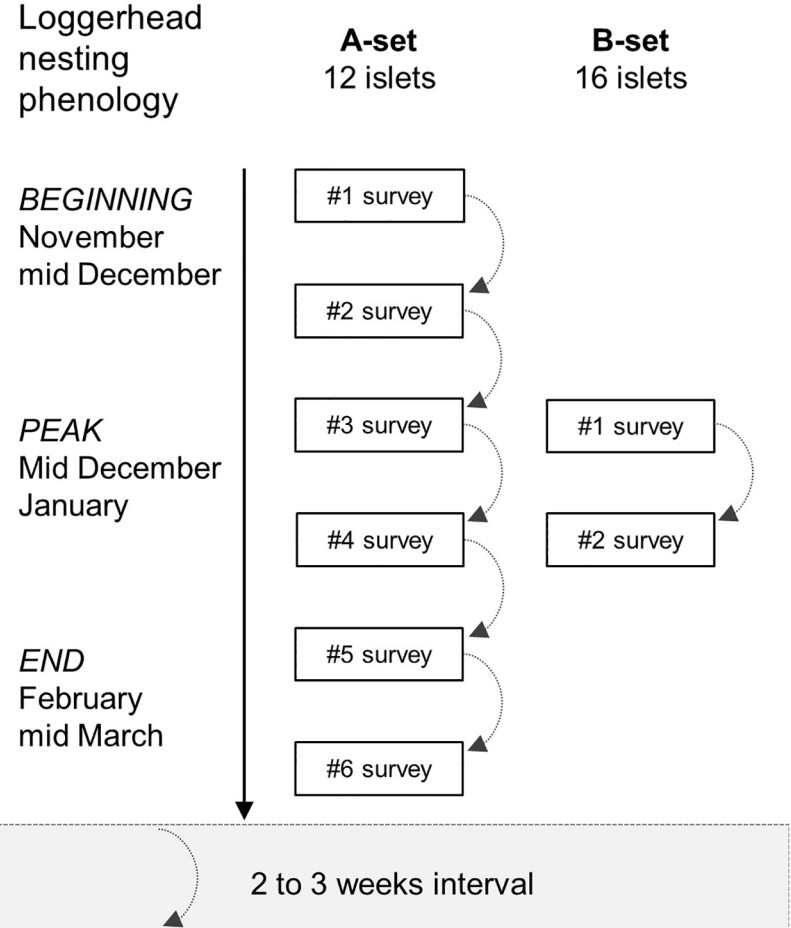

**Fig 2. Monitoring strategy of the two sets of surveyed islets in the 'Grand Lagon Sud' rookery.**

presumably used by the same nesting population. Setting a well-adapted monitoring scheme is challenging, since no preconized protocol reflects the entire complexity of the present situation [11]. Given the logistic impossibility to monitor each islet with high intensity throughout the nesting season, the initial selection of 28 islets has been divided into two distinct sets, respectively designated as A- and B-set, that will differ in the total seasonal number of survey occurrences (Fig 2 and Table 1).

The A-set is composed of a panel of 12 islets showing various levels of nesting intensities. It benefits the highest monitoring effort, with a minimum of six survey events throughout the nesting season. Surveys were conducted with a two to three weeks interval in between them in order to (1) take into account the loggerhead inter-nesting interval, which varies between 12 to 25 days based on available sources [20, 25], (2) efficiently assess the primary nesting phenology parameters, prerequisite key knowledge to the use of statistical methods to quantify total seasonal number of nests, and (3) provide valuable and whenever possible exhaustive counts from one survey event to another, considering the decrease of the nest detection over time. The 16 remaining islets, composing the B-set, were monitored only twice during the season, with a two to three weeks interval at the peak of the nesting season [26], *i.e.* from mid-December to mid-January [20]. It would later benefit from the nesting phenology knowledge generated

through the A-set survey protocol to produce robust estimations of the seasonal total number of nests on those sites. This monitoring strategy enables a reduced survey coverage while maintaining solid estimates of seasonal total nest counts [11].

## 2.3 Modeling nesting abundance

**2.3.1 Nest tracks persistence over time.**   Although nesting trends can be detected from nest counts, an understanding of what proportion of the true abundance these counts represent is key to more accurately assess demographic progressions [11]. Without an estimate of detection probability, *i.e.* the proportion of nests recorded on a given survey event, a simple nest count is not as reliable an index of population status or trends [27]. In order to assess this parameter with high precision, we have developed an empiric protocol that allows us to determine the average duration for which a nest track remains detectable for an observer. It is based on the fact that the marks generated by females during nesting activities persist in the sand for a given period of time, which may vary according to weather conditions and to their wind, rain and wave exposure [9]. Nests were marked on the spot with wooden poles on the first detection, then systematically prospected in following survey events. A status of their detection state was set by the observers, from good (coded as 1) to poor (uncertain or no detection, coded as 0) (S3 File).

These data have been analyzed with a generalized linear mixed model using Penalized Quasi-Likelihood with a binomial distribution and a logit link using the function glmmPQL of the 'MASS' R package [28]. Age of the track in days and season (centered to December) were included as fixed factors and ID of the track has been included as a random factor with an autocorrelation structure of order 1 to take into account that the detection state (1 or 0) at time $t$ is dependent on the detection state at time $t$-$\Delta t$. If the track was not detectable at time $t$, there are more chance that it is still not detectable at time $t$-$\Delta t$, except for false negative or positive cases. Confidence interval is not available for GLMM-AR1 model and we used 9999 bootstraps to estimate the uncertainty of the predictions.

**2.3.2 Model for nesting seasonality.**   Collected data were processed in a multiple-step pipe. The aim is to convert partial fieldwork data into an estimate of total seasonal nesting intensity through the establishment of the nesting phenology parameters and aggregation over multiple seasons. Assuming that $t$ is an ordinal date (October 1 is 1 and September 30 in the following year is 365 or 366) and that $N_t$ is the observed number of nests for this date, the number of nests deposited per night is modeled using the following set of equation:

$$n_t = \begin{cases} t < B \to \text{PMin} \times \text{Max} \\ t \in [B, P-F] \to ((1 + \cos(\pi(P-F-t)(P-F-B)))/2)(\text{Max} - \text{PMin} \times \text{Max}) + \text{PMin} \times \text{Max} \\ t \in [P-F, P+F \to \text{Max} \\ t \in [P+F, E] \to ((1 + \cos(\pi(t-P+F)(E-P+F)))/2)(\text{Max} - \text{PMin} \times \text{Max}) + \text{PMin} \times \text{Max} \\ t > E \to \text{PMin} \times \text{Max} \end{cases} \quad (1)$$

The model requires at most seven parameters, all of which have direct biological interpretations. A graphic explanation of these terms is available in Girondot [29].

$B$ and $E$ are the ordinal dates for the start and end of the nesting season.

$P$ is the ordinal date for the peak of the nesting season.

$F$ is half of the number of days around $P$ for which the curve flattens out.

*Max* is the mean number of nests at the peak of the nesting season.

*PMin* is the mean nightly nest numbers relative to *Max* before and after the nesting season.

The nesting season is described in segments, and all segments form one continuous function. The nesting season is defined as the interval $[B, E]$. If $F$ is equal to 0, no flat portion is observed. Rather than fitting $B$ and $E$, it is more convenient to fit LengthB = $P–B$ and LengthE = $E–P$ with LengthB > 0 and LengthE > 0 to ensure that $B < P < E$. The parameters $B$, $E$, $P$, $F$, LengthB, and LengthE are hereafter defined as shape parameters, and $PMin$ and $Max$ as scale parameters.

In such a situation when several nesting seasons are analyzed, it is also possible to implement a year effect for Peak ($P$) and/or for LengthB (length of the nesting season from the beginning to the peak) and LengthE (length of the nesting season from the peak to the end). Four categories of models were then fitted depending on the year effects for Peak and/or for LengthB and LengthE. These are defined as the Peak-Global or Peak-Year and Length-Global or Length-Year.

For the seasonality model, parameter fitting was performed using maximum likelihood with negative binomial (NB) daily nest distribution with values produced by Eq (1) ($m = n_t$) as theoretical values and the observed counts ($x = N_t$) as observations. In ecology, NB distribution is used to describe the distribution of an organism while taking into account the mean number of individuals $m$ and an aggregation parameter $k$ [30]. The probability mass function of NB distribution is:

$$NB(x; m; k) = Pr(X = x) = \frac{\Gamma(k + x)}{x!\,\Gamma(k)} \left(\frac{m}{m + k}\right)^x \left(\frac{k}{m + k}\right)^k, \ m > 0, \ k > 0 \qquad (2)$$

When the mean number of nests is low during all the season, a Poissonian distribution rather than a NB distribution can be used [31]. Poissonian distribution is a special case of NB distribution when $k$ is +Inf. When $N_t$ is equal to 0, it is replaced with $10^{-9}$ as the negative binomial and the Poissonian models are not defined for $m = 0$.

When the count ($N_t$) represents the exact sum of activity during several nights, the probability mass function of the sum of NB is used [32]. When the count represents the minimum number of activities because of the track loss in time, the likelihood is the integral of the probability mass function of NB for all the values between $N_t$ and the maximal number of nests $N_{max}$. $N_{max}$ is determined taking into account the probability of detection of a body pit after a known period on the beach (see section 3.1).

Nesting seasonality was modeled following the Girondot phenological model [29, 31], using the R package 'Phenology' available in the Comprehensive R Archive Network (https://cran.rproject.org) [40]. This model can be applied to any proxy of nesting such as nest or track counts. Scripts are provided in the following GitHub repository https://github.com/Marc-Girondot/NC2024/.

**2.3.3 Interannual spatial and temporal trends.** A model is used to estimate the number of nest tracks for any of the considered islets that has not been surveyed on a given season, based on the number of tracks recorded on this islet during other monitored seasons, the relative frequency of tracks on the different sites and the total number of tracks for each nesting season. Let the total theoretical number of nest tracks be $T_i$ for season $i$ in the entire region where $K$ islets were monitored during a range of $Y$ years.

The distribution of the nests across the different sites is defined by the proportion $p_j$ of $T_i$ nests in the $j$ beach. For a total of $K$ islets, $K–1$ parameters $p$ are necessary due to the relation:

$$\sum_{j=1}^{K} p_j = 1 \qquad (3)$$

The values $p_j$ can be modeled as constant ($K$–1 parameters) or first (2.$K$–2 parameters) or second (3.$K$–3 parameters) order as a function of time to represent situations with changes in the relative use of the different nesting sites. The expected number of nests for year $i$ in the beach $j$ is then:

$$E_{i,j} = T_i * p_j \qquad (4)$$

Let $N_{i,j}$ be an observed number of nests with a standard deviation of $S_{i,j}$. The distribution of $N_{i,j}$ may be close to a Gaussian distribution when the number of nests and monitoring coverage are high, but it can also be a positive skew when the number of nests or monitoring coverage is low. For this reason, a gamma distribution was used to model the data; the gamma distribution is always positive and can show a positive skew when the standard deviation is high compared to the mean. The fit of the parameters was then completed using maximum likelihood with a gamma distribution. For the final estimate, the expected number of nests $E_{i,j}$ was only used when no observation was available; in other situations, the number of nests fitted using the phenology model was preferred.

The log likelihood of the observations is the sum of the log likelihood for each observation $N_{i,j}$ within the gamma model. This model was implemented using the R package 'Phenology' [40].

**2.3.4 Strategy for parameter fitting.**   The parameters used in the models to implement the nesting phenology (see section 2.3.2) and to estimate the annual number of nests for islets that were not monitored over a given season (see section 2.3.3) were fitted using the same statistical approach. First, the parameters are fitted using maximum likelihood, and then the models are selected using the Akaike Information Criterion (AIC) [33] and Akaike weight [34]. In short, AIC evaluates the quality of the fit that penalizes for overfitting too many parameters, while the Akaike weight gives the relative support of the different models, *i.e.* the probability for each model being the best one.

Finally, the distribution of parameters was assessed using the Metropolis-Hastings algorithm, which is a Markov chain Monte Carlo method for obtaining a sequence of random samples from a probability distribution [35, 36]. The adaptive proposal distribution [37] as implemented in R package 'HelpersMG' [38] ensures that the acceptance rate is close to 0.234, which is the optimal acceptance rate [39]. A total of 20,000 iterations was run. Priors were all uniform with a range of proposals large enough to ensure that it did not constrain the limits of the parameters. From the 20,000 sets of parameters, we calculated the posterior predictive mean and standard deviation of the statistics of interest.

The adjustments were performed using the R package 'Phenology' [40]. Comparisons between the observed and modeled values were based on the adjusted coefficient of determination [41].

## 2.4 Ethics statement

All implemented protocols were approved by the relevant ethics and legal committee of the Environmental Department of the South province (DDDT), the local legal authority regarding endangered species, under permits #3553-2016/ARR/DENV, 3385-2017/ARR/DENV, 4276–2018/ARR/DENV, 2837-2019/ARR/DENV and 3245-2020/ARR/DDDT. This study implied no human experimentations; therefore, no informed consent was required.

## 3 Results

Over the 6 nesting seasons covered by this study (2017–2018 to 2022–2023), 51 monitoring field trips were conducted, for a total effort of 98 days and over 300 hours of beach survey.

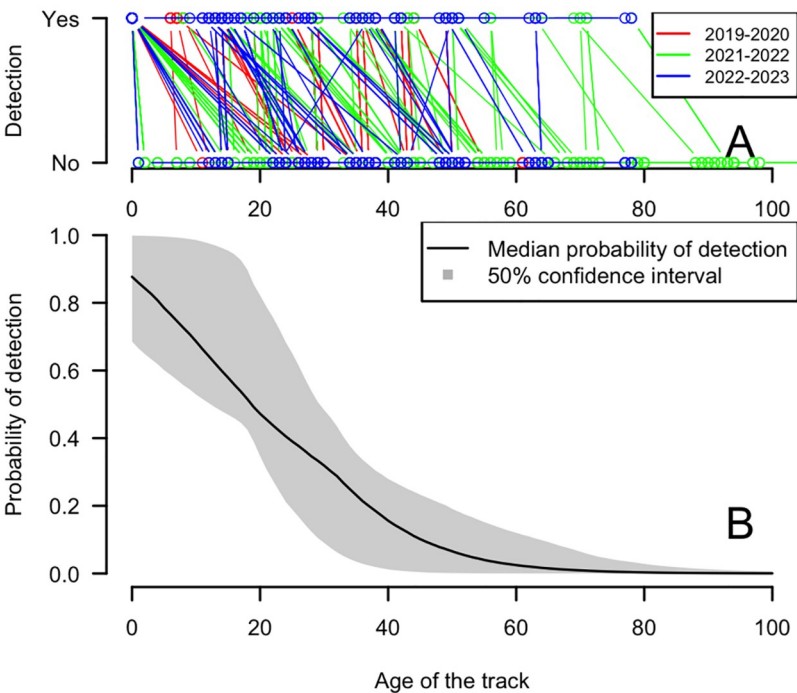

**Fig 3. Evolution of nest detection over time.** (A) Empirical data of detection of tracks according to their age for 3 seasons. (B) Model of probability of detection based on logistic function.

This protocol led to the census of 2186 tracks, including 1105 nests based on the presence of body pits over the 28 prospected islets.

## 3.1 Nest detection probability

The loggerhead turtle nest tracks persistence estimated on the 'GLS' rookery was based on the survey of n = 286 nests over the 2019–20 to 2022–23 period. Empirical data are presented in Fig 3A. The confidence interval of predictions for each season overlaps nearly fully (not shown) and it confirms that season effect is not present in this dataset (Wald t-test for Season effect, Fisher combined p-value = 0.11). Autocorrelation structure is very high with Phi = 0.47. It can be seen in Fig 3A that very few false positive or negative detections are present in the dataset (a false positive or negative occurs when a non-detected track was seen again in the next field session). The fitted detection probability is shown in Fig 3B including its confidence interval based on $10^4$ bootstrapping estimates. The fitted parameters of the logit model are: Intercept = 1.82, Age = -0.08 which can be converted into $S$ and $A$ of the detection probability $d_a$:

$$d_a = 1/(1 + exp((-1/(4.S))(A - a)))$$ (5)

With $S$ = -(1/(4*Age)) = 3.04 and $A$ = 4*S*Intercept = 22.23. The half-time life of a body pit on these beaches is 22.23 days (95% confidence interval (CI) from 1 to 71 days, 50% CI from 5 to 29).

### 3.2 Nesting phenology of the loggerhead turtle in the 'Grand Lagon Sud' rookery

We fitted the 'GLS' 2017–2018 to 2022–2023 time series with Peak, LengthB, LengthE, Pmin and Max nesting phenology parameters using the R package phenology [40]. Flat parameter was not retained as it was fitted always close to 0. A Poisson distribution was preferred to a Negative Binomial since the mean number of nests per night is very low [31] and if NB was used, $k$ fitted value was very large (>1000). We also tested different combinations of parameters to fit the more accurate phenological model and implement the possible seasonal for peak (P) and/or for LengthB and Length E (LBLE), based on AICc [33]. The selected model has a different seasonal peak date but a similar nesting season length for all 6-time series (Table 2).

The fitted dates for the nesting seasonality in the 'GLS' rookery over the study period are shown in Table 3. The season starts in late October, and stretches for over 4 months into the austral summer. The peak of the activity occurs in late December/early January.

### 3.3 Assessment of the overall seasonal number of loggerhead turtle nests on the 'GLS' rookery

For each prospected islets, time series of nest counts has been fitted using the phenological parameters *LengthB*, *LengthE* and *Peak* of the corresponding nesting season as fixed parameters, with only the *Max* parameter being specific to each islet time series [13, 31]. We obtained the posterior predictive mean number of nests during the season, then we estimated the number of nesting activities for the islets not monitored during a season. Ultimately, an overall estimate of the nesting intensity at the rookery-scale was generated, as well as its distribution based on 50,000 iterations using a Markov chain built with Monte Carlo sampling (Table 3). The overall seasonal number of nests in the 'GLS' rookery ranged from 303 (95% CI = 232–405) during the 2020–2021 season to 532 (95% CI = 377–715) during the 2017–2018 season, with a mean seasonal intensity of 437 nests (95% CI = 328–582) over the 2017–2018 to 2022–2023 period. Seasonal number of nests and 95% CI for each islet are available in S1 Table.

## 4 Discussion

We have estimated the overall nesting activity of loggerhead turtles in the remote, scattered and hard-to-reach 'GLS' rookery for the first time. Based on an adapted monitoring protocol, we were able to highlight an important nesting intensity, ranging annually from 232 to 402 nests for the lowest limit of the 95% CI (Table 3). This result is quite exceptional considering both the precarious conservation status of the South Pacific subpopulation and available

**Table 2. Model selection for the phenology of the number of nesting activities in the 'GLS' rookery during the 2017–2018 to 2022–2023 period.**

| Peak & LBLE model | AICc | ΔAICc | Akaike weight |
|---|---|---|---|
| Peak common + LBLE common | 1640.46 | 5.95 | 0.05 |
| **Peak seasonal + LBLE common** | **1634.51** | **0.00** | **0.95** |
| Peak common + LBLE seasonal | 1646.28 | 11.77 | 0.00 |
| Peak seasonal + LBLE seasonal | 1649.32 | 14.80 | 0.00 |

If the Peak (*P*) and LengthB/LengthE (LBLE) are indicated as 'common', then this indicates that a single set of values is used for the 'GLS' rookery, otherwise a different set of values is used for each season, indicated as 'seasonal'. The selected model is indicated in bold. AICc: Akaike Information Criterion corrected for small sample sizes. ΔAICc is the difference in AICc from the best-performing model.

**Table 3. Fitted nesting phenological parameters in the 'GLS' rookery and median seasonal number of nests with the 95% Credible Interval (CI).**

| Nesting season | 2017–18 | 2018–19 | 2019–20 | 2020–21 | 2021–22 | 2022–23 |
|---|---|---|---|---|---|---|
| **Phenology parameters** | | | | | | |
| Beginning | 02 Nov. | 04 Nov. | 31 Oct. | 30 Oct. | 27 Oct. | 17 Oct. |
| Peak | 30 Dec. | 01 Jan. | 28 Dec. | 27 Dec. | 24 Dec. | 14 Dec. |
| End | 27 Mar. | 29 Mar. | 24 Mar. | 24 Mar. | 21 Mar. | 11 Mar. |
| Length (days) | 145 | 145 | 145 | 145 | 145 | 145 |
| **Seasonal number of nests** | | | | | | |
| Lower 95% CI | 377 | 402 | 312 | 232 | 347 | 299 |
| median | 532 | 531 | 455 | 303 | 437 | 362 |
| Upper 95% CI | 715 | 709 | 661 | 405 | 553 | 450 |

Fit for the selected nesting phenology model: date of the Beginning, Peak, End and Length of the loggerhead turtle nesting season.

territorial knowledge prior to this study [17, 20]. The mean number of nests 437 (95% CI = 328–582) surpasses that of the estimated number of loggerhead turtle nesting activities in New Caledonia.

The assessment of nest detection dynamics over time on the islets of the 'GLS' rookery has allowed us to feed the models more accurately. The resulting nesting phenology parameters estimated are consistent with local reports in New Caledonia, with a season duration of 4.8 months, starting in late October/early November and with a peak date occurring in late December/early January [20].

Monitoring protocol replication potential and conservation implications.

The management of marine turtle species, with ocean-wide distributions and philopatric behaviors, can be facilitated by better understanding their reproductive population trends [9]. The lack of standardized data series in New Caledonia has been highlighted by the IUCN in 2015 as a source of uncertainty regarding the assessment of the South Pacific subpopulation [17]. A strong emphasis has thus been given to the need of developing robust estimates of nesting intensity on high potential sites, such as the 'Grand Lagon Sud' area. Comprehensive monitoring on this important zone to the loggerhead turtle has been hindered by the financial and technical difficulties to adequately survey the numerous and distant coral islets inventoried in the area. Therefore, we have compiled a 6-year time-series to adapt a monitoring and analytic protocol in order to estimate the overall number of nesting activities. Resulting knowledge has allowed us to identify the 'GLS' rookery as the first nesting area in terms of number of nests for the Critically Endangered loggerhead turtle South Pacific subpopulation in New Caledonia. This study also establishes the preliminary basis in the perspective of assessing the demographic trends, which requires several years, if not decades, of survey effort due to the marine turtle life history traits [42]. The only conclusion that can be drawn at this stage is that the population is not facing an imminent crash or an extremely rapid growth.

The late identification of this major area to the reproduction of loggerhead turtles in New Caledonia is very likely to be attributed to the lack of efficient tools to assess the nesting intensity in this geographical context. Considering that other important nesting aggregations had been identified in New Caledonia, especially in the north-western lagoon where both the number of nesting islets and the monitoring constraints are very similar to that found in the 'GLS' area, it is reasonable to assume that the importance of other nesting rookeries has been overlooked. The assessment of such areas could prove useful to highlight innovative conservation actions to undertake for the more efficient preservation of this emblematic species.

## 5 Conclusions

In the light of this study, the importance of New Caledonia to the South Pacific loggerhead turtle subpopulation nesting population has been refined, with a mean estimation of 20% (14–39%) of all reproductive females breeding and nesting within the territory. An important knowledge gap is still to be filled considering other high-potential insular rookeries in the South Pacific region, such as the north-western lagoon of New Caledonia that could prove to be as important as the 'GLS' rookery. The implementation of this protocol could substantially improve both regional knowledge and management policies of sea turtle populations, as the monitoring scheme is highly replicable to other similar geographies and/or to other sea turtle species.

## Supporting information

**S1 File. DB_nestcounts_mission_20240129.**
(PDF)

**S2 File. DB_nestcounts_total_20240129.**
(PDF)

**S3 File. DB_nestdetectability_total_20240129.**
(PDF)

**S1 Table. Seasonal number of nests and 95%CI for each considered islet.**
(PDF)

## Acknowledgments

We would like to thank the South Province of New Caledonia for their substantial contribution to the study, by providing precious technical and fieldwork support and administrative authorizations to seasonally visit restricted-access sites in the 'Grand Lagon Sud' area. We extend our gratitude to all WWF-France volunteers, the French National Research Institute for Sustainable Development (IRD) and the Coastal Observatory of Geological Services of New Caledonia (OBLIC) members who contributed to field missions. Finally, we would like to thank all the local tribes in which geographies this study has taken place.

## Author Contributions

**Conceptualization:** Hugo Bourgogne, Marc Oremus, Marc Girondot.

**Data curation:** Hugo Bourgogne, Marc Oremus, Morgan Mangeas, Marc Girondot.

**Formal analysis:** Hugo Bourgogne, Morgan Mangeas, Marc Girondot.

**Funding acquisition:** Hugo Bourgogne, Marc Oremus.

**Investigation:** Hugo Bourgogne, Marc Oremus.

**Methodology:** Hugo Bourgogne, Marc Oremus, Marc Girondot.

**Project administration:** Marc Oremus, Morgan Mangeas.

**Resources:** Marc Oremus.

**Software:** Marc Girondot.

**Supervision:** Marc Oremus, Morgan Mangeas, Eric Vidal, Marc Girondot.

**Validation:** Hugo Bourgogne, Marc Oremus, Morgan Mangeas, Marc Girondot.

**Writing – original draft:** Hugo Bourgogne, Marc Girondot.

**Writing – review & editing:** Hugo Bourgogne, Marc Oremus, Morgan Mangeas, Eric Vidal, Marc Girondot.

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
