## [Decision Letter · Decision Letter 0]

1 Apr 2024

PONE-D-24-06149Innovative monitoring scheme at remote, scattered nesting aggregation reveals a major loggerhead turtle rookery in New Caledonia, South PacificPLOS ONE

Dear Dr. Bourgogne,

Thank you for submitting your manuscript to PLOS ONE. After careful consideration, we feel that it has merit but does not fully meet PLOS ONE’s publication criteria as it currently stands. Therefore, we invite you to submit a revised version of the manuscript that addresses the points raised during the review process.

 In this study, a new survey protocol and statistical model-based approach for predicting sea turtle nest numbers in remote, difficult-to-reach areas of New Caledonia are presented. The manuscript is well written. The manuscript has been reviewed by two reviewers and based on these revisions, the authors need to edit or correct their manuscript accordingly.

We look forward to receiving your revised manuscript.

Kind regards,

Murtada D. Naser

Academic Editor

PLOS ONE

Reviewers' comments:

Reviewer's Responses to Questions

**Comments to the Author**

1. Is the manuscript technically sound, and do the data support the conclusions?

Reviewer #1: Yes

Reviewer #2: Yes

2. Has the statistical analysis been performed appropriately and rigorously? 

Reviewer #1: Yes

Reviewer #2: Yes

3. Have the authors made all data underlying the findings in their manuscript fully available?

Reviewer #1: Yes

Reviewer #2: Yes

4. Is the manuscript presented in an intelligible fashion and written in standard English?

Reviewer #1: Yes

Reviewer #2: No

5. Review Comments to the Author

Reviewer #1: Short title: The proposed short title is not short. Suggestion: Innovative monitoring scheme at remote loggerhead rookeries

The MS is written nicely and clearly, the proposed protocols and analyses are well-founded, and the references are adequate. Statistical analyses are rigorous and well-explained.

Data is made available in the supporting information provided. The data might be used to test the models presented and help to support the findings. Suggestion: For non-mathematician readers, R scripts providing examples of the code used in the analyses may facilitate the use for other researchers. Scripts may be provided as supplementary data or in a GitHub repository.

In section 2.3. I suggest rearranging the text so that in this section's parts, the last paragraph describes the software used for the analyses. For example, in the section “2.3.2 Model for nesting seasonality “, I suggest specifically mentioning it at the end (last paragraph) for example: “Nesting seasonality was modelled the Girondot phenological model (29,30) using R package “phenology” (and cite reference 40 here)”.

In this way, all sections follow the same structure: explaining the model and software used if needed.

In section 2.3.3. Did the authors use a specific R package? Mention it (see above comment)

Reviewer #2: Dear Authors,

I have read your manuscript titled "Innovative monitoring scheme at remote, scattered nesting aggregation reveals a

major loggerhead turtle rookery in New Caledonia, South Pacific" with interest. This paper presents a new method for estimating sea turtle nest counts in isolated, hard-to-reach locations in New Caledonia based on a new survey protocol and the use of statistical models.

While I find your approach fascinating, I do, however, have some questions and suggestions.

Lines 33-34. You mentioned that this rookery hosts a large nesting colony, with a mean annual estimate of 437 nests. How many females does this represent? Is the number similar to that of major rookeries in Australia?

Line 135. What is the average size of these islets? How much time does a person require to survey each islet thoroughly? This information should be part of the protocol as it is part of the survey effort.

Lines 194 - 198. You did an excellent job explaining the parameters of the model, but I think a graphic explanation might be very helpful.

Lines 229 - 231. This statement was not clear to me. When you say "islets not surveyed", do you refer to those within the 28 considered in this study or additional ones? Additionally, if the islets are of different sizes, and the available nesting habitat is different on each, wouldn't this affect your estimates? I assume the larger islets would have more nests than the small ones. How do you estimate the number of nests if you did not factor in the size of the islets?

Line 253. This sentence was a bit confusing. Could you please rephrase it to make it clearer?

Throughout the text, I found some typos and grammatical errors that need to be corrected. As someone whose native language is not English, I strongly suggest a full review by someone who can help correct the grammar better than I can.

Best wishes.

6. PLOS authors have the option to publish the peer review history of their article (what does this mean?). If published, this will include your full peer review and any attached files.

Reviewer #1: **Yes: **Eduardo J. Belda

Reviewer #2: No

---

## [Author Response · Author response to Decision Letter 0]

16 May 2024

Response to reviewers

PLOS ONE manuscript number: PONE-D-24-06149

Article type: Research article

Noumea, May 15, 2024

Dear Reviewers,

First, on behalf of all the co-authors, I would like to address our gratitude to you for having accepted to review the paper entitled ‘Innovative monitoring scheme at remote, scattered nesting aggregation major loggerhead turtle rookery in New Caledonia, South Pacific’.

I have read your comments and suggestions with interest, and I am very humbled for the interest you have manifested for our work. I have found positive, constructive and well-assigned questions and reviews, and I have tried to answer it and/or take it into account the best I could.

Please find here the rebuttal letter that will respond each point raised with a personal answer and the ensuing modifications to the manuscript (MS).

All co-authors and I remain at your disposal for any further information.

Kind regards,

Hugo Bourgogne.

 

Responses to reviewer #1 Edouardo J. Belda (EB)

- EB Short title: The proposed short title is not short. Suggestion: Innovative monitoring scheme at remote loggerhead rookeries

HB Thank you for this more appropriate suggestion. We have opted for the alternative ‘sea turtle rookeries’ since the monitoring scheme can actually be extended to any sea turtle species. 

MS lines 6-7: ‘Innovative monitoring scheme at remote sea turtle rookeries’

- EB Suggestion: For non-mathematician readers, R scripts providing examples of the code used in the analyses may facilitate the use for other researchers.

HB Done. As indicated in the core manuscript, all scripts are provided in the GitHub repository https://github.com/Marc-Girondot/NC2024/

- EB In section 2.3. I suggest rearranging the text so that in this section's parts, the last paragraph describes the software used for the analyses. For example, in the section “2.3.2 Model for nesting seasonality “, I suggest specifically mentioning it at the end (last paragraph) for example: “Nesting seasonality was modelled the Girondot phenological model (29,30) using R package “phenology” (and cite reference 40 here)”. In this way, all sections follow the same structure: explaining the model and software used if needed.

HB The models used to implement the nesting phenology (section 2.3.2), to estimate the number of nests for islets that were not monitored over a given season (section 2.3.3) or to fit the parameters in either case (section 2.3.4) were all performed using the same R package ‘phenology’. Initially, it was hence mentioned at the end of the last section (lines 267 to 270). We have added the mention for each section as you suggested.

MS Removal lines 185-187: ‘Nesting seasonality was modeled following the Girondot phenological model [29,30]. This model can be applied to any proxy of nesting such as nest or track counts.’

Insertion after line 227: ‘Nesting seasonality was modeled following the Girondot phenological model [29,30], using the R package ‘Phenology’ available in the Comprehensive R Archive Network (https://cran.rproject.org) [40]. This model can be applied to any proxy of nesting such as nest or track counts. Scripts are provided in the following GitHub repository https://github.com/Marc-Girondot/NC2024/.’

Insertion line 251: 'This model was implemented using the R package ‘Phenology’ [40].’

Removal lines 267-268: ‘available in the Comprehensive R Archive Network (https://cran.r-project.org) that implements these models’.

- EB In section 2.3.3. Did the authors use a specific R package? Mention it (see above comment)

HB Same response than in previous comment. The mention has been added in the appropriate section.

Response to reviewer #2 (R2)

- R2 Lines 33-34. You mentioned that this rookery hosts a large nesting colony, with a mean annual estimate of 437 nests. How many females does this represent? Is the number similar to that of major rookeries in Australia?

HB Extrapolating a number of reproductive females from the annual number of nests is a hazardous exercise as the number of nests laid by females may significantly vary depending on the individual, the rookery or the season. However, it is safe to presume that this nesting intensity implies the presence of hundreds of females, with a total stock that could number 400-500 individuals. Limpus & Casale (2015) reported the annual presence of 500-2000 reproductive females in eastern Australia. Considering that female loggerhead turtles breed every 1-7 years, it could be assumed that the total eastern Australian reproductive females’ stock could number 1500-6000 individuals. In the present state of our knowledge, the GLS area could therefore aggregate 7-23% of the reproductive females of the South Pacific subpopulation. 

- R2 Line 135. What is the average size of these islets? How much time does a person require to survey each islet thoroughly? This information should be part of the protocol as it is part of the survey effort.

HB The average perimeter length of the islets is 813.2 m, and the average duration for one person to complete an islet patrol is 39.5 min.

MS Insertion line 128: ‘The average perimeter length of the islets is 813.2 m (+ 310.0), and the average duration for one person to complete an islet patrol is 39.5 min (+ 3.4).’

- R2 Lines 194 - 198. You did an excellent job explaining the parameters of the model, but I think a graphic explanation might be very helpful.

HB A graphic explanation is indeed available in the reference Girondot, 2010. We suggest mentioning it and referring the reader to this reference.

MS Insertion line 193: ‘A graphic explanation of these terms is available in Girondot [29].’

- R2 Lines 229 - 231. This statement was not clear to me. When you say "islets not surveyed", do you refer to those within the 28 considered in this study or additional ones?

HB The designation ‘islets not surveyed’ indeed referred to those identified within the 28 considered in this study.

MS Removal line 229: ‘for an islet that is not surveyed, based on’

Insertion line 229: ‘for any of the considered islets that has not been surveyed on a given season, based on the number of tracks recorded on this islet during other monitored seasons,’

- R2 Lines 229 - 231. Additionally, if the islets are of different sizes, and the available nesting habitat is different on each, wouldn't this affect your estimates? I assume the larger islets would have more nests than the small ones. How do you estimate the number of nests if you did not factor in the size of the islets?

HB All the beaches of the selected islets were entirely monitored at least over two distinct nesting seasons. Consequently, the estimates we obtain already take into account the size of the islet because the linear of coast depends on the size of the islet. Thus, larger islets have more beach linear and host more nesting than smaller ones. In the total estimate, the representation of large islets will be greater than that of small islets.

- R2 Line 253. This sentence was a bit confusing. Could you please rephrase it to make it clearer?

HB Done

MS Line 253-255: ‘The parameters used in the models to implement the nesting phenology (see section 2.3.2) and to estimate the annual number of nests for islets that were not monitored over a given season (see section 2.3.3) were fitted using the same statistical approach.’

- R2 Throughout the text, I found some typos and grammatical errors that need to be corrected. As someone whose native language is not English, I strongly suggest a full review by someone who can help correct the grammar better than I can.

HB Thank you for alerting us on this issue and for having contributed to improve the language of the manuscript through your numerous annotations, which were accepted. We have gone through the text again, and I hope this has helped perfecting the manuscript.

---

## [Editor Report · Decision Letter 1]

21 May 2024

Innovative monitoring scheme adapted to remote, scattered nesting aggregation reveals a major loggerhead turtle rookery in New Caledonia, South Pacific

PONE-D-24-06149R1

Dear Dr. Bourgogne,

We’re pleased to inform you that your manuscript has been judged scientifically suitable for publication and will be formally accepted for publication once it meets all outstanding technical requirements.

Kind regards,

Murtada D. Naser

Academic Editor

PLOS ONE
---

## [Editor Report · Acceptance letter]

27 May 2024

PONE-D-24-06149R1 

PLOS ONE

Dear Dr. Bourgogne, 

I'm pleased to inform you that your manuscript has been deemed suitable for publication in PLOS ONE. Congratulations! Your manuscript is now being handed over to our production team.

Kind regards, 

on behalf of

Dr. Murtada D. Naser 

Academic Editor

PLOS ONE